# Influence of Microplastics on Manifestations of Experimental Chronic Colitis

**DOI:** 10.3390/toxics13080701

**Published:** 2025-08-21

**Authors:** Natalia Zolotova, Maria Silina, Dzhuliia Dzhalilova, Ivan Tsvetkov, Nikolai Fokichev, Olga Makarova

**Affiliations:** 1Department of Immunomorphology of Inflammation, Avtsyn Research Institute of Human Morphology of Federal State Budgetary Scientific Institution “Petrovsky National Research Centre of Surgery”, 117418 Moscow, Russia; marusyasilina99@yandex.ru (M.S.); juliajal93@mail.ru (D.D.); davedm66@gmail.com (I.T.); fokichev.1993@mail.ru (N.F.); makarov.olga2013@yandex.ru (O.M.); 2Department of Histology, Petrovsky Medical University, 119435 Moscow, Russia; 3Faculty of Biology and Biotechnology, HSE University, 117418 Moscow, Russia

**Keywords:** microplastics, chronic colitis, ulcer, goblet cells, mucins, enteroendocrine cells, claudins, proliferation, apoptosis, cytokines

## Abstract

Environmental pollution with microplastics (MPs) can have a negative impact on human health. Certain findings point to the relationship between MP and the development of inflammatory bowel diseases (IBD). We investigated the effect of MP consumption on the severity of chronic colitis in male C57BL/6 mice. The MP effect was modeled by drinking water consumption with a suspension of 5 μm PS particles at a concentration of 10 mg/L replacement for 12 weeks. Chronic colitis was induced by three seven-day cycles of 1% DSS consumption (starting from the 8th, 29th and 50th days of the experiment). We investigated inflammatory infiltration, the goblet cell volume fraction and the highly sulfated and neutral mucins content in them, the endocrine cell number, the ulcerative-inflammatory process prevalence, changes in the gene’s expression encoding tight junction proteins, glycocalyx components proapoptotic factor Bax and proliferation marker Mki67 in the colon, and TNFα, IL-1β, IL-6 and IL-10 cytokines content in the serum. In healthy mice, MP did not cause pathological changes in the colon; however, indirect data indicate an increase in colon permeability. In chronic colitis, MP leads to higher prevalence of all pathological changes in general, and ulcers in particular, in a greater number of crypt abscesses and enteroendocrine cells. MP consumption leads to a more severe chronic colitis course.

## 1. Introduction

Due to the growing plastics production, a global environmental problem is the pollution of the environment by small plastic particles, less than 5 mm in size, characterized as microplastics (MPs). MP particles could be detected in the air, soil, water, living organisms, and also in the human body [1,2]. Therefore, an important goal is to determine the potential MP harm to human health.

MP enters the human body mainly with water and food, and their first target is the gastrointestinal tract. MP probably affects the development of inflammatory bowel diseases (IBD) [3,4,5]. According to Yan et al. (2022), the MP content in the feces of patients with IBD is higher than in healthy people, and there is a positive correlation between the MP concentration in feces and the disease severity [6].

IBD is a group of chronic relapsing inflammatory gastrointestinal tract disorders. The two major IBD types are Crohn’s disease, characterized by skip lesions and transmural inflammation that can affect any part of the gastrointestinal tract, and ulcerative colitis (UC), with inflammation limited to mucosa and submucosa of the colon and rectum. Their etiology remains unclear. It is assumed that many factors play a role in the development of IBD: genetic predisposition, dietary and lifestyle characteristics, intestinal microflora disorders, stress, and environmental factors [3,4,5,7]. In 2019, there were approximately 4.9 million IBD cases worldwide, with China and the USA demonstrating the highest number of cases (66.9 and 245.3 cases per 100,000 people, respectively) [8].

During the last five years, experimental studies on mice were conducted in order to determine the effects of MP on the intestine. It was demonstrated that in the colon of healthy mice, MPs cause oxidative stress, increased permeability, mucosa infiltration by immune cells, increased pro-inflammatory cytokines production, and a decrease in the goblet cell number and mucus production. MP consumption leads to changes in proliferation, apoptosis, epithelial cells differentiation, tight junction and glycocalyx components expression, membrane transport, intracellular signaling pathways, metabolome, and intestinal microbiota composition [9]. According to our previous studies, six weeks of 5 μm polystyrene microparticles consumption with water at a dose of 2.3 mg/kg/day led to an increase in the number of endocrine cells, an increase in the content of highly sulfated mucins in goblet cells, and an increase in the number of cells in the lamina propria [10]. Although no obvious pathological changes are observed in the colon when exposed to MP, oxidative stress, local pro-inflammatory reactions and barrier function disorders can contribute to the colitis development caused by other factors.

At the moment, only seven articles are published describing the MP effect on the experimental colitis severity in mice. According to most studies, MP leads to a more pronounced course of both acute and chronic colitis [11,12,13,14,15]. However, Schwarzfischer et al. (2022) did not reveal the MP effect on either acute or chronic colitis courses [16]. In our previous work, we demonstrated that polystyrene particles with a diameter of 5 μm at a dose of 2.3 mg/kg/day for six weeks of consumption lead to a more pronounced acute colitis course in mice, which is characterized by a greater ulcerative and inflammatory process prevalence and a decrease in the neutral mucins in the goblet cell level on the seventh day from the colitis induction start (acute colitis was modeled by administration of 1% dextran sulfate sodium (DSS) to the drinkers from the thirty-sixth to the fortieth day of the experiment) [10].

Since the IBD etiology remains unclear, diagnostic and treatment methods that are currently available are not effective enough. Clinical manifestations of IBD are non-specific and include diarrhea, rectal bleeding, fatigue and weight loss. Diagnosis is also complicated by the fact that approximately 25–40% of people with IBD have extraintestinal manifestations such as arthritis, axial spondyloarthritis, uveitis, erythema nodosum and primary sclerosing cholangitis. From the first appearance of IBD symptoms to diagnosis, an average of two months to six months passes, and in some cases, diagnosis can take up to eight years [17]. Thus, by the time the diagnosis is established, the inflammatory process in the intestines has already reached the chronic phase. The course of these diseases, characterized by phases of relapse and remission, is unpredictable. Today, IBD is a medically incurable disease and the goal of therapy is to achieve stable remission [7]. However, treatment often only results in the disappearance of clinical symptoms. According to Rosenberg et al. (2013), at an endoscopy 45% of patients in clinical remission had endoscopic inflammation [18]. Therefore, the most adequate model of human IBD is experimental chronic colitis in the clinical remission phase [19,20].

With regard to the above, the aim of this work is to evaluate the MP consumption effect on the chronic experimental colitis course in mice.

The most widely used and highly reproducible animal model for studying IBD is DSS-induced colitis. This model allows for inducing both acute and chronic colitis of varying severity. The DSS-induced colitis model is more representative of UC than Crohn’s disease [19,20]. Therefore, we used a DSS-induced colitis model and, when choosing parameters to assess the chronic experimental colitis severity, we relied on studies of UC in humans.

The main indicator of the severity of UC in humans is histological changes in the colon [21]. In the DSS-induced colitis, to evaluate the inflammation severity in the colon we commonly used a histological score, a semi-quantitative measure derived from ulcers, crypt abscesses and inflammatory infiltration [22,23,24,25].

An important role in UC development is played by disturbances in the colonic epithelial barrier. The first component of this barrier is the mucus produced by goblet cells. Mucus forms two layers; the first is an inner dense layer, which is permeable to bacteria and particles larger than 0.5 μm. The second layer is loose and populated by commensal microflora. The main structural component of colon mucus is highly glycosylated glycoprotein, mucin MUC2. The terminal mucins carbohydrate groups can be modified by sulfate or sialic acid residues (acidic mucins) or unmodified (neutral mucins). It was demonstrated that in UC, there is a significant reduction in the mucus layer thickness and the number of goblet cells, increased amounts of sialomucins, and decreased mucins sulfation. The degree of these changes correlated with the severity of colitis. In knockout mice for the mucin 2 gene (Muc2^−/−^), the mucus layer is absent and spontaneous colitis develops, while in heterozygous Muc^2+/−^ mice, DSS-induced colitis is significantly more severe than in wild-type animals. The next epithelial barrier component is the glycocalyx, a carbohydrate-rich layer of transmembrane molecules on the colonic epithelial cells apical surface. In the colon, the primary structural components are membrane-associated mucins. In humans, the major mucins of the colonic glycocalyx are MUC3, MUC12, and MUC17, which share a high degree of structural similarity and are encoded by a single gene (Muc3) in mice. Data on the role of these mucins in the UC development are contradictory. The human colon glycocalyx contains a significant amount of MUC13 mucin. Mice with a knockout of the mucin 13 gene (Muc13^−/−^) develop more severe DSS-induced colitis than wild-type animals. The abundance of MUC1 in the colonic glycocalyx is not high, but this molecule contains several important signaling sequences. In UC patients, MUC1 expression increases and anti-MUC1 antibodies are detected in the blood. In knockout mice for the mucin 1 gene (Muc1^−/−^), the mucus layer is thicker, and these animals develop DSS-induced colitis with significantly reduced severity in comparison to wild-type animals. The colonic mucus is lined with a single layer of columnar epithelium. All epithelial cells are formed from stem cells located at the bottom of the crypts. The main types of differentiated epithelial cells are colonocytes, goblet cells, and enteroendocrine cells, each with specialized functions. Colonocytes are the primary absorptive cells, responsible for water and electrolyte absorption. Goblet cells secrete mucus, which lubricates and protects the epithelial lining. Enteroendocrine cells produce and secrete hormones that regulate various digestive processes. The predominant enteroendocrine cell type in the colon is Ec-cells (enterochromaffin cells) that secrete serotonin. UC is characterized by a decrease in the goblet cell number, which correlates with the UC severity; during remission, the goblet cell number increases. The endocrine cell number in the colon and the serotonin content in the tissue decrease during UC exacerbation, and increase during remission and in the long-term disease course. Epithelial cells are connected by junctional complexes: desmosomes, adhesive junctions, and tight junctions. For the colon barrier function, tight junctions are the most important. They are impermeable to macromolecules and selectively permeable to some ions. Tight junctions are formed by intersecting chains of transmembrane proteins that interact with proteins of the neighboring cell and form a network of point connections between membranes. Claudins are the main tight junction structural components. In the patients with UC, the level of claudin 2 protein increases, and the content of mucins 4 and 7 decreases. Knockout of claudins 2, 4, and 7 genes in mice results in more severe DSS colitis or development of spontaneous colitis [26,27,28].

Therefore, for analysis we selected colon parameters such as the severity of ulcers and inflammation, changes in the goblet and endocrine cells content, mucin sulfation, transmembrane mucins expression, claudins, proliferation, and apoptosis markers. To characterize the systemic manifestations of colitis, we assessed cytokine levels in the blood.

## 2. Materials and Methods

### 2.1. Animals

The study was performed on 32 adult male mice C57BL/6 obtained from the branch “Stolbovaya” of the Federal State Budgetary Institution of Science’s “Scientific Center for Biomedical Technologies of the Federal Medical and Biological Agency”, Moscow, Russia. At the time of receipt, the mice were 1.5 months old, and a body weight of 16–20 g. Technicians, unaware of the experimental plan, randomly assigned the animals to 4 cages. The mice were kept 8 animals per cage in an open system at a room temperature with natural light, free access to water and compound feed. Each cage was randomly assigned to one of 4 experimental groups. The experimental unit was a single animal.

### 2.2. Experimental Groups

Four experimental groups of 8 animals were formed (Figure 1): (1) C—control group, animals consumed distilled water for 12 weeks, (2) MP—consumption of microplastics for 12 weeks, (3) CC—chronic colitis induction over 11 weeks, (4) MP+CC—chronic colitis against the background of MP consumption.

When replacing solutions and suspensions in drinking bowls, the approximate liquid volume consumed by the mice was estimated from their residue. The animals were also weighed once a week. The sample size was determined according to similar studies [12,13,14]. The animals were taken out of the experiment on day 84 using cervical dislocation under ether anesthesia. There were no excluded animals or data points. All experiments were performed simultaneously, cages with animals were located in the same room, next to each other. Animal care technicians were blinded to the experimental design.

### 2.3. Microplastic Consumption Model

According to the literature data, MP exposure is mostly often modeled on male C57BL/6 mice, using PS particles with a 0.5 or 5 μm diameter, doses from 0.2 to 20 mg/kg/day, and 4–6 weeks exposure [9].

As a MP, we used microparticles based on polystyrene (PS) (79633, Supelco, Bellefonte, Pennsylvania, USA): aqueous suspension, 0% cross-linked, concentration 10% (solids), particle size 5 μm std dev < 0.1 μm, coeff var < 2%, density 1.05 g/cm^3^.

Before use, the stock MP suspension was thoroughly mixed in a shaker and treated with ultrasound to prevent particle aggregation, and then added to a drinking bowl with distilled water to a final concentration of 10 mg/L. Glass drinking bowls and distilled water were used to avoid foreign plastic particles entering the water. The drinking bowl was shaken several times a day to prevent particles from settling. The plastic type, particle size, and slurry concentration were consistent with previous studies [10,29,30].

### 2.4. Chronic Colitis Model

Chronic colitis induced by 3 cycles of DSS (Dextran sulfate sodium salt, Mr ~40,000, AppliChem, Darmstadt, Germany) treatment [19,20]. DSS was added into the drinking bowl for 7 days starting from the 8th, 29th and 50th days of the experiment to a final concentration of 10 g/L (1% solution). The intervals between DSS exposure cycles were two weeks. Four weeks passed from the last DSS cycle to the end of the experiment.

In the MP+CC group, a suspension of polystyrene microparticles was added to the drinking bowl throughout the experiment to a final concentration of 10 mg/L, and starting from the 8th, 29th and 50th experimental days for 7 days, additional DSS was added to a final concentration of 10 g/L. As in the other studies on the effects of MP on colitis severity, no additional substances were added to stabilize the MP suspension [11,12,13,14,15].

### 2.5. Evaluation of Fluid Intake

A glass drinking bowl was filled with 125 mL of distilled water and we added 12.5 μm of a stock MP suspension (10% suspension, 100 mg/mL). The drinking bowl was closed and shaken thoroughly by hand. The drinking bowl was installed in the cage. Two to three times every day, the drinking bowl was removed from the cage, shaken by hand to prevent the MP particle sedimentation, and installed back in the cages. Twice a week, the suspension in the drinking bowls was replaced. The drinking bowls were removed from the cages, the remaining liquid was poured into a measuring flask, and the volume of the remainder was measured (between 10 and 40 mL remained). After measuring the volume, the remaining liquid was poured out, the drinking bowls were thoroughly washed, and a new suspension of MP was prepared. During manipulations with the drinkers, several drops of water were lost. According to our rough estimates, the losses were less than 5 mL per drinker per week, and we neglected these losses. From the initial volume of liquid (125 mL), the volume of the remainder (10–40 mL) was subtracted, added for the number of mice in the cage (8 animals), and divided by the number of days of liquid consumption from the drinker (3–4 days); thus, the liquid consumption of the animals was calculated. The consumption of distilled water (group C) and sodium dextran sulfate solution (group CC) was calculated similarly.

### 2.6. Histological Study of the Colon

#### 2.6.1. Preparation of Histological Specimens

In DSS-induced colitis, the most pronounced morphological changes are observed in the distal colon [20]. Therefore, a histological study of the distal colon was performed. The colon was straightened on a filter, opened along the mesentery, and cleared of contents, and the distal section (approximately 1.5 cm) was placed in 10% neutral drilled formalin (Biovitrum, Saint Petersburgб Russia). It was fixed for 24 h, washed with tap water, and stored in 70% ethanol for further studies. After histological processing, placed in histomix and longitudinal sections 5 μm thick were made. Histological sections were stained with hematoxylin and eosin, Alcian blue pH 1.0 (highly sulfated mucins), a PAS reaction was performed (neutral mucins), and immunohistochemical staining was performed with antibodies to chromogranin A (endocrine cells).

#### 2.6.2. Prevalence of the Pathological Process

Usually, a semi-quantitative assessment of the histological score is used to assess the severity of colitis. However, we decided to use a more accurate quantitative method [31]. Hematoxylin and eosin-stained sections of mouse colon from the CC and MP+CC groups were scanned using a slide scanner at a magnification of 100. In the QuPath-0.5.1 program, the length of sections without pathological changes with inflammatory infiltration and with ulcers was measured along the lamina muscularis mucosae. The percentage of the section length with pathological changes was calculated. Also, the number of crypt-abscesses on the section was counted.

#### 2.6.3. Severity of Inflammatory Infiltration

Photographs of hematoxylin and eosin-stained sections were obtained at a magnification of 320 in 2 fields of view (areas with longitudinally cut crypts and without ulcers were selected). In the PhotoM 1.21 program, mucosa lamina propria areas were outlined, their area was determined, and the number of nuclei in these fields was counted. The number of cells per 1 μm^2^ of mucosa lamina propria was calculated.

#### 2.6.4. Goblet Cell Content and Mucus Properties

Terminal carbohydrate groups in mucins can be unmodified or modified by acids. Unmodified carbohydrate groups are detected using the PAS reaction: periodic acid oxidizes vicinal diol groups into dialdehydes, these dialdehydes then react with Schiff’s reagent, a colorless solution. Alcian blue is a basic dye that binds to acidic groups at pH 1.0. Alcian blue binds only to strong acids and stains highly sulfated mucins. Photographs of sections stained with Alcian blue pH 1.0 and after the PAS reaction were obtained at a magnification of 200 in 2 fields of view (areas with longitudinally cut crypts and without ulcers were selected). The goblet cell area and the brightness of their staining were determined. In the ImageJ (Version 1.54p) program, binarization of the images was performed, setting the threshold so that only goblet cells were highlighted. The colonic wall fields from the muscular plate to the lumen were outlined, and its area and the area of the goblet cells on it were determined. The volume fraction of the goblet cells was calculated as the ratio of their area to the area of the mucosa (PAS reaction). The average brightness of the dots of the goblet cells and the background (the area of the image without tissue) was measured. The optical density of the goblet cells was calculated as the common average brightness of background points to the average brightness of goblet cell points ratio logarithm. The higher the optical density, the higher the content of highly sulfated (Alcian blue) or neutral (PAS reaction) mucins in the goblet cells was.

#### 2.6.5. Enteroendocrine Cell Content

To detect endocrine cells, immunohistochemical staining was performed with antibodies to chromogranin A (Rabbit-anti-mouse polyclonal anti-Chromogranin A antibody, ab15160, Abcam Inc, Cambridge, UK). Secondary antibodies with a peroxidase label Donkey anti-Rabbit IgG (H+L) Highly Cross-Adsorbed Secondary Antibody, HRP (A16035, Invitrogen, Carlsbad, USA) were used. Photographs of sections were obtained at a magnification of 100 in 2 view fields. The mucosa area and number of chromogranin A-positive cells on it were counted. The number of chromogranin A-positive cells per 1 mm^2^ of mucosa area was calculated.

### 2.7. Real-Time PCR

A fragment of the medial colon 1 cm long was placed in the IntactRNA fixative (BC031, Evrogen, Moscow, Russia) for subsequent RNA extraction and gene expression analysis.

The qRT-PCR method was used to evaluate changes in the gene’s expression encoding tight junction proteins—claudins *Cldn2*, *Cldn4*, *Cldn7*, and glycocalyx components—mucins *Muc1*, *Muc3*, *Muc13*, proapoptotic factor *Bax* and proliferation marker *Mki67* in the medial colon. The RNA Solo kit (Eurogen, Moscow, Russia) was used to isolate RNA, and the MMLV RT kit (Eurogen, Moscow, Russia) was used to synthesize cDNA. The expression levels of the studied genes were evaluated relative to the expression level of β-actin *Actb* mRNA. PCR was performed using a ready-made 5X qPCRmix-HS SYBR mixture (Eurogen, Russia) with oligonucleotides at final concentrations of 0.2–0.4 μM (Appendix A) on a DTprime device (DNA-Technology, Moscow, Russia). Primers for PCR were selected using the online program primer-BLAST in accordance with generally accepted requirements, synthesized in “Eurogen” (Russia).

The relative concentration of mRNA was calculated using the Formula (1):(1)$$\frac{{\left[A\right]}_{0}}{{\left[B\right]}_{0}}=E^{∆C(T)}$$
where [A]_0_ is the initial concentration of gene mRNA in the PCR mixture; [B]_0_ is the initial concentration of *Actb* mRNA in the PCR mixture; E is the reaction efficiency (taken as 1.98); and ΔC(T) is the difference between the threshold cycles of *Actb* and the target gene.

### 2.8. ELISA

Blood was collected from the neck vessels. The content of TNFα, IL-1β, IL-6 and IL-10 cytokines in the blood serum was determined using ELISA Kit (EM0183, EM0109, EM0121, EM0100, FineTest, Wuhan, China).

### 2.9. Statistics

Statistical processing of the obtained data was performed using the STATISTICA 6.0 program (StatSoft, Inc., Tulsa, OK, USA). Nonparametric statistical methods were used due to the small sample size and abnormal distribution of parameter values (*x*^2^ criterion). The samples were described by the median and interquartile ranges Me (25%; 75%). The Mann–Whitney test was used, with Bonferroni correction for multiple comparisons, to compare the groups. Differences were considered statistically significant at an error probability of *p* < 0.05 when comparing 2 groups and at *p* < 0.0085 when comparing 4 groups. If 0.0085 < *p* < 0.05 when comparing 4 groups, we described the changes as “trends”.

## 3. Results

The animals were weighed once a week. The weight of the mice did not differ between the groups, did not change significantly during the experiment and was averagely 24.4 ± 2.5 g (Mean ± SD). After solutions and suspensions were replaced, the remaining liquid volume was measured. Liquid consumption did not differ between the groups and was 3.6 ± 0.5 mL/mouse/day. PS particles at the 10 mg/L concentration and DSS at the 10 g/L (1%) concentration were added to the drinkers. Accordingly, the PS dose was approximately 1.48 mg/kg/day, and the DSS dose was 1.48 g/kg/day. The diameter of the polystyrene spheres was 5 μm, and the density was 1.05 g/cm^3^; therefore, the dose in terms of particles was 2.17 × 107 particles/kg/day.

Histological examination of the animal’s distal colon from groups C and MP revealed no pathological changes. The epithelial lining was preserved throughout the section, the crypts were deep with a narrow lumen, and a small number of cellular elements were evenly distributed in the mucosa lamina propria and submucosa: fibroblasts, lymphocytes, histiocytes, and single neutrophils (Figure 2).

In groups CC and MP+CC, the morphological picture in the colon was mosaic. Some zones did not differ from the control. There were areas with pronounced inflammatory infiltration, mainly from lymphocytes and plasma cells. Violations of the crypt histoarchitectonics were revealed: deformed crypts with an expanded lumen, sometimes branching, and some crypts were not located parallel to each other. Narrow and extensive epithelialized ulcers reaching the mucosa muscular plate were detected. Single crypt abscesses were encountered. In the MP+CC group, adenocarcinoma was detected in one animal (Figure 3).

In the mice of the MP group, compared with the control group, the content of cells in the mucosa lamina propria statistically significantly increased. The chromogranin A-positive endocrine cell number increased. There was a tendency toward a decrease in the goblet cell volume fraction and the content of highly sulfated mucins in them, but the differences did not reach statistical significance (Figure 4 and Figure 5). In the CC and MP+CC groups, we assessed morphological changes in the zones outside the ulcers, but with the pronounced inflammatory changes. Compared with the control group, the number of cells in the mucosa lamina propria in both groups increased. The number of chromogranin A-positive endocrine cells did not differ from the control. The volume fraction of goblet cells in the CC group statistically significantly decreased, and in the MP+CC group there was a tendency to decrease. The highly sulfated mucins content in the MP+CC group statistically significantly decreased, and in the CC group there was a tendency to decrease. The neutral mucins content did not change.

Between the MP+CC and CC groups, a difference was found only in the content of endocrine cells; it was higher in animals consuming MP (Figure 4 and Figure 5). MP consumption led to an increase in the ulcer prevalence of and all pathological changes, as well as the crypt abscesses number of chronic colitis (Figure 6, Table 1).

In the MP group, in comparison to the control, there was a tendency for an increase in *Cldn2* mRNA expression. In the CC group, compared to the control, there was a statistically significant decrease in *Muc1* mRNA expression. In the MP+CC group, compared to the control, there was a tendency for a decrease in *Muc1* and *Cldn2* mRNA expression. Compared to the CC group, in the MP+CC group there was a tendency for a higher level of *Cldn7* mRNA and a lower mRNA of *Cldn2* (Table 2).

In the blood serum of animals from the MP group, compared with the control, a decrease in the content of TNFα and IL-1β was observed. The levels of IL-6 and IL-10 did not change. In chronic colitis, in animals receiving and not receiving MP, the content of TNFα, IL-1β, IL-6 and IL-10 in the blood serum did not differ from the control values (Figure 7).

## 4. Discussion

### 4.1. Microplastic Consumption

Accordingly, the PS dose was approximately 1.48 mg/kg/day, and the DSS dose was 1.48 g/kg/day. The polystyrene spheres’ diameter was 5 μm, and the density was 1.05 g/cm^3^; therefore, the dose in terms of particles was 2.17 × 10^7^ particles/kg/day. Existing methods for MP identifying and determining its concentration in biological samples of humans, animals, and food products are not standardized and not accurate. Evaluations of human MP consumption are mostly indirect and based on estimates of MP content in water and food. According to various estimates, human MP intake is from 0.2 to 10.2 mg/kg/day or from 2 to 1.1 × 10^6^ particles/kg/day [9]. The MP dose in our study corresponds to human MP intake assessment of Senathirajah et al. (2021): 0.2–10 mg/kg/day [32]. In addition, it should be taken into account that the food passage rate through the digestive system in mice is approximately 10 times higher than in humans. In addition, mice have a significantly higher metabolic rate than humans. Therefore, the dose used in our study is adequate for human MP exposure, at least in plastic-polluted regions.

We did not investigate the penetration of MP across the intestinal epithelial barrier. In humans it is assumed that MP absorption may only be limited to ≤0.3% [33]. Liang et al. (2021) [34] treated mice with a single oral gavage of fluorescent PS particles 0.05, 0.5 or 5 μm-diameter in a dose of 250 mg/kg and analyzed its distribution in organs after 24 h. The total bioavailability for 0.05, 0.5 or 5 μm particles were 6.16, 1.53 and 0.46%, respectively. Particles 0.05 and 0.5 μm in size were detected in the spleen, kidneys, heart, liver, lungs, blood, testis and epididymis, brain and thighbone, some individual signs in the muscles and breastbone, some 0.05 μm particles in the ovaries and uterus. The particles 5 μm in size were found in the blood only. The total accumulations of 0.05, 0.5 or 5 μm particles in the intestine were 11.41, 13.66 and 3.84%, respectively. The particles 5 μm in size accumulated primarily in the stomach and large intestine [34]. In the study of Liu et al. (2022) [12], healthy mice and mice with acute colitis were exposed to 5 μm-diameter fluorescent PS particles at concentrations of 500 μg/L for 28 days and then fluorescent MP distribution in mice was assessed by using a small animal living imaging system. In the healthy mice, a small amount of fluorescence was distributed in the abdomen, whereas a 13 times stronger fluorescence intensity was detected in the intestinal tract of mice with acute colitis [12]. Thus, PS particles 5 μm-diameter are almost not absorbed in the digestive system, and they accumulate in the large intestine. In colitis their intestinal absorption increases, probably due to the formation of ulcers and increased gut permeability.

### 4.2. Effect of MP on the Colon of Healthy Mice

Consumption of MP particles for 12 weeks did not lead to the development of pathological changes in the colon, which is consistent with the literature data [10,35,36]. A number of studies have described the effects of MP, such as an increase in the distance between crypts [22,23,25,37,38], an increase or decrease in the depth of crypts [14,23,37,39,40,41,42], an increase in the content of lymphocytes [22,23,24,35,40,43,44,45,46], a decrease in the number of goblet cells [38,40,42,43] in the colon mucosa. However, pronounced inflammatory infiltration and development of ulcers were not detected in any study. Therefore, MP alone, even with prolonged exposure, is not able to induce colitis in healthy animals without additional factors.

We noted a tendency for the goblet cell volume fraction to decrease in the colon when exposed to MP, which was confirmed by the literature data [10,14,23,24,25,34,35,37,42,44,45,46,47,48]. Goblet cells produce mucus that covers the epithelial lining of the mucosa. In the large intestine, mucus forms two layers: internal and external. The internal layer of mucus is dense, is not removed by aspiration, and is impermeable to bacteria and particles larger than 0.5 μm. In the human distal colon, it has a thickness of about 200–300 μm, and in mice, about 50 μm. The outer mucus layer is looser, easily washed off and abundantly populated by commensal microorganisms, formed as a result of partial degradation and loosening of the mucin network of the inner mucus layer [49]. The terminal carbohydrate groups of mucins can be modified by sulfuric or sialic acid residues (acid mucins), or unmodified (neutral mucins). In comparison to neutral mucins, acid mucins (especially sulfomucins) are more resistant to enzymatic cleavage by both bacterial glycosidases and proteases of the body itself [50,51]. Data on changes in the content of acidic and neutral mucins in goblet cells upon exposure to MP are contradictory. In our previous study of the MP exposure for six weeks’ effects, the content of highly sulfated mucins in goblet cells increased, while neutral mucins did not change [10]. According to other studies, the content of sulfated mucins did not change [52] or decreased [37] when mice consumed MP, while neutral mucins increased [52]. In the current study, we noted a tendency towards a decrease in the highly sulfated mucins content. The data inconsistency is due to the use of different types and sizes of MP particles, different doses, and exposure duration. A decrease in the goblet cell volume fraction and the proportion of acidic mucins in them indicates the colonic mucosal barrier weakening.

The next colonic epithelial barrier component after the mucus layer is the glycocalyx, the main structural components of which are transmembrane mucins. These mucins protrude 200–1500 nm above the surface of the epithelial cell [53]. The main mucins of the human colon glycocalyx are MUC3, MUC12, and MUC17. They are structurally very similar and in mice they correspond to one gene: *Muc3*. The glycocalyx of the human colon also contains a significant amount of MUC13. Its extracellular domain is significantly shorter than that of other transmembrane mucins, and it probably forms the next line of defense, acting as a bacterium that has passed through the layer of long transmembrane mucins sensor [49]. The content of MUC1 in the glycocalyx of the colon is low; however, it is involved in many signaling pathways. It interacts with Src kinases, proapoptotic kinase ABL, and scaffold protein GRB2 (Growth factor Receptor Bound protein 2), causing the activation of MAPK (Mitogen-Activated Protein Kinase) pathways, heat shock proteins HSP70 and HSP90, β-catenin and others. In UC, its expression increases, and anti-MUC1 antibodies are detected in the blood of patients [54]. According to the literature, exposure to MP reduces the content of glycoprotein and mRNA *Muc1* [34,37,44,46,47,48], mRNA *Muc3* [34,47] and *Muc13* [34]. We did not reveal any changes in the expression of these mucins. It is possible that with longer exposure to MP (twelve weeks in our work in comparison to four to six weeks in other studies), the body adapts and the expression of transmembrane mucins normalizes.

Behind the glycocalyx are the actual epithelial cells of the colon, connected to each other by a complex of intercellular contacts. Paracellular transport through the epithelial layer is regulated by tight junctions. They form the lateral adjacent epithelial cells membrane’s maximum convergence zone. Tight junctions are formed by transmembrane proteins intersecting chains that interact with adjacent cell proteins and form membrane point connections networks. The main structural components of tight junctions are claudins. Claudins 1, 2, 3, 4, 5, 7, 8, 10, 12, 15, and 23 are expressed in the large intestine [55]. Single Cell RNA biopsies sequencing from patients with IBD demonstrated an increase in the expression of claudins 1, 2, and 18, and a decrease in claudins 3, 4, 5, 7, 8, and 12 [56]. According to Liang et al. (2021), MP causes an increase in the claudins *Cldn3*, *Cldn4*, and *Cldn7*’s mRNA expression in the mouse colons [34]. We revealed a tendency towards an increase in the *Cldn2* mRNA expression. DSS-induced colitis in mice with claudin-2 deficiency (*Cldn2^−/−^*) is more severe [57], and in transgenic mice with claudin-2 overexpression it is milder [58] than in wild-type animals. The tendency towards an increase in *Cldn2* expression observed by us is probably an adaptive response to damage to the epithelial barrier.

The main types of differentiated colon epithelial cells are border colonocytes, which absorb water, ions and nutrients; goblet cells, which produce mucus; and endocrine cells, that secrete regulating gastrointestinal tract hormones. The predominant type of endocrine cells in the colon are EC-cells, which produce serotonin. We detected endocrine cells using antibodies to chromogranin A, a high-molecular acidic glycoprotein that acts as an osmotic stabilizer of secretory vesicles [59]. When exposed to MP, the number of chromogranin A-positive endocrine cells in the distal colon increased, which is consistent with the literature data [10,39]. An increase in the number of chromogranin A-positive endocrine cells indicates an increase in serotonin production. Serotonin in the gastrointestinal tract stimulates the secretion of mucus and digestive enzymes, suppresses the formation of hydrochloric acid, inhibits the absorption of water and electrolytes in the intestine, and enhances its motility. In addition, serotonin affects the immune system: during acute inflammation, serotonin promotes the mast cells, eosinophils, dendritic cells, and neutrophils attraction; modulates the dendritic cells differentiation; increases the pro-inflammatory cytokines production; and enhances the macrophages phagocytic activity and the NK cells cytotoxic activity [60]. The changes we observed are probably a protective adaptive response aimed, on the one hand, at eliminating MP from the intestine by increasing mucus secretion and reducing water absorption, and on the other hand, at enhancing local immune protection.

In our work, an increase in the cell content in the colonic mucosa lamina propria was observed. Our results are consistent with the literature data on an increase in the lymphocytes number in the colonic mucosa when exposed to MP [22,23,24,25,38,40,43,44,45,46]. Furthermore, Djouina et al. (2022) revealed an increase in the ratio of polymorphonuclear neutrophils and a decrease in the proportion of anti-inflammatory macrophages [39], and Huang et al. (2023) revealed an increase in the ratio of CD11b+Ly6C+ pro-inflammatory monocytes and CD11b+Ly6G+ neutrophils [38] in the colonic mucosa in mice receiving MP. Apparently, these changes are due to the immune cell migration from the blood to the colon tissue in response to increased luminal antigens translocation due to epithelial barrier disruption. According to the literature data, mice that consumed MP had higher fluorochrome-labeled dextran concentrations with a molecular weight of 4 kDa in the blood in comparison to intact animals, and 70 kDa when administered orally [23,25,34,47,48,61,62]. Exposure to MP increased the bacteria number in the mesenteric lymph nodes, liver, and spleen [25,48].

We revealed a decrease in the pro-inflammatory cytokines TNFα, IL-1β level in the blood serum of the mice receiving PS particles with a diameter of 5 μm for 12 weeks, at a dose of 1.48 mg/kg/day. The literature data on the MP effect on the cytokines levels in the mice blood are contradictory. According to Li et al. (2020), in animals receiving PE particles of 10–150 μm in size at doses of 0.3, 3 or 30 mg/kg/day for 35 days, the IL-1α in the blood level increased; at a low dose, the IL-2 level decreased and IL-6 increased; and at a high dose, the IL-5 level decreased and IL-9 increased [22]. According to the study by Liu et al. (2022), 28 days of exposure to 0.1 and 5 μm PS particles at a 0.05 mg/kg/day dose leads to an increase in the TNFα and IL-1β levels in the blood [63]. Luo et al. (2022) did not reveal changes in the TNFα, IL-1β, IL-6 and IL-10 levels after 21 days of 5 μm PS particles at doses of 20 and 200 mg/kg/day consumption [13]. In the work of Zheng et al. (2021), the TNFα, IL-1β and IFNγ levels increased in the blood of mice receiving 5 μm PS particles at a dose of 0.1 mg/kg/day for 28 days [11]. Zha et al. (2024) demonstrated that consumption of 5 μm PS particles at a dose of 22 mg/kg/day for 35 days leads to a decrease in the IL-4 in the blood level, while the IL-2 and IL-6 levels do not change [36]. At this particular study stage, it was not possible to fully explain the decrease in blood levels of pro-inflammatory cytokines TNFα and IL-1β after MP consumption. This issue requires additional research.

According to the literature, MP influences apoptosis and colonic cell proliferation. When exposed to MP in the colon, there was an increase in the mRNA intestinal stem cell markers: *Lgr5*, *Bmi1* and *Olfm4* expression, an increase in the proliferation markers *c-Myc* and *Pcna* volume fraction in the colon [14,39], an increase in the TUNEL+ dying cell number, the cleaved caspase-9 level, cleaved caspase-3 and pro-apoptotic factor Bax [34,44,52] levels, and a decrease in the anti-apoptotic factor Bcl-2 level [44] were demonstrated. However, we did not detect changes in the mRNA expression of the pro-apoptotic factor *Bax* and the proliferation marker *Mki67*, which may be associated with adaptation to long-term exposure to MP.

Thus, MP causes colon epithelial barrier disruption; its permeability increases and translocation of luminal antigens and microorganisms into the blood and other internal organs is enhanced. However, the organism’s adaptive reactions prevent the pathological process development.

### 4.3. Chronic Colitis Model

In the groups of chronic colitis with and without MP, we observed superficial epithelialized ulcers; pronounced lymphocytic-plasmacytic inflammatory infiltration; crypt architecture disruption and crypt abscesses; and a decrease in the goblet cell volume fraction and the content of sulfomucins in them, which corresponds to the literature data on chronic DSS-induced colitis in mice and moderate activity of UC in humans [64,65,66].

In one animal with chronic colitis with MP consumption, adenocarcinoma of the colon was detected. According to the literature, in chronic colitis induced by four cycles of DSS, colon tumors develop in approximately 10% of mice [67]. In people with a long-term course of the disease, the risk of developing colorectal cancer is high [68].

In both groups with chronic colitis, a decrease in the transmembrane mucin *Muc1* mRNA expression in the colon was observed. In humans, the MUC1 level in the intestine increases during severe disease in the exacerbation phase [69]. In knockout mice for the mucin 1 gene (*Muc1*^−/−^), the mucus layer is thicker, and they develop DSS-induced colitis of significantly less severity than in wild-type animals [70]. The decrease in Muc1 expression is probably an adaptive response.

Although the chronic DSS-induced colitis model is widely used, there is very little information in the literature on changes in the components of the epithelial barrier. In our previous studies, we showed that in mice with chronic colitis induced by one cycle of 1% DSS solution for five days, on the twenty-eighth day of the experiment, compared with the control, the content of sulfomucins in goblet cells in the distal colon was statistically significantly reduced. The volume fraction of goblet cells, the content of neutral mucins in them, and the number of chromogranin A-positive enteroendocrine cells did not differ from the control. In the medial colon, the expression of *Muc1* and *Cldn2* mRNA increased, *Muc13* decreased, and the expression of *Muc3* and *Cldn4* mRNA did not differ from the control group [71,72].

The pro-inflammatory cytokines TNFα, IL-1β, IL-6 and anti-inflammatory IL-10 levels in animals with chronic colitis did not differ from the control values. From the end of the last DSS cycle to the end of the experiment, four weeks passed and all ulcers in the colon were epithelialized, so there were no systemic signs of inflammation. Thus, over the course of our study, we successfully reproduced an adequate chronic colitis model to UC clinical remission in humans.

### 4.4. Effect of MP on the Chronic Colitis Severity

In the distal colon, the number of cells in the mucosa lamina propria, the goblet cell volume fraction and the neutral and sulfomucins content in them did not differ between the animal groups with chronic colitis that received and did not receive MP. This is probably due to the mucous membrane areas chosen with the same severity of pathological changes for analysis: areas with pronounced inflammatory infiltration, but without ulcers. According to our previous studies, in acute colitis (colitis for 7 days, MP for 42 days), mice receiving MP had a lower content of neutral mucins in goblet cells in the colon in comparison to animals with acute colitis that did not receive MP, and the number of cells in the mucosa lamina propria did not differ [10]. According to Luo et al. (2022), at later stages of colitis (colitis for 21 days, MP for 21 days), when exposed to MP, the goblet cell volume fraction was lower [13]. The contradiction between our data and the literature is probably associated with the colitis to chronic phase transition and the exposure to MP’s (colitis for 77 days, MP for 84 days) longer duration.

In chronic colitis without MP, the chromogranin A-positive endocrine cells content in the colon did not differ from the control values, which is consistent with our previous studies [72]. In the animals with chronic colitis against the background of the MP consumption group, this indicator was higher in comparison to chronic colitis without MP and the control group and corresponded to the values in the group of MP consumption without colitis. According to our previous study, MP effects studies on the course of acute colitis, the numbers of endocrine cells in animals receiving and not receiving MP did not differ and were higher than in the control group [10]. Both DSS and MP probably directly or indirectly have a stimulating effect on serotonin production in the colon. In the chronic colitis group, four weeks passed from the last exposure to DSS to the animals euthanized, and the hormone production level and the cell number had time to normalize. In the chronic colitis group, against the MP consumption background the endocrine cells content remained high, due to the continued exposure to polystyrene microparticles until the experiment finished.

According to the literature data, in acute colitis (7 and 21 days), the blood of animals receiving MP had higher TNF-α, IL-1β, IL-10 and IFN-γ levels [11,13]. In chronic colitis (63 days), MP did not affect the TNF-α, IL-β, IL-6 and IL-10 blood levels [13], which is consistent with our data.

We did not investigate any MP effect on the transmembrane mucins expression in the colon during chronic colitis. There was no literature data revealed towards this point.

Compared to colitis without MP, in the animals with colitis against the background of the MP consumption group, the expression of *Cldn2* mRNA was slightly lower and *Cldn7* was higher. Transgenic mice with an increased expression of claudin 2 are more resistant to DSS-induced colitis in comparison to wild-type mice. They demonstrate increased epithelial cells proliferative activity, decreased DSS-induced death, increased levels of regulatory T-lymphocytes in the colon, and a decreased immune response to DSS activation [73]. The observed decrease in *Cldn2* upon exposure to MP indicates a more pronounced damage to the epithelial barrier. In claudin 7 knockout mice (*Cldn7^−/−^*), ulceration of the mucosa, epithelial cell detachment, and inflammation in the colon are observed [74]. In our study, increased *Cldn7* expression is likely a compensatory response to decreased *Cldn2*.

We investigated that of all pathological changes prevalence, the ulcers and the number of crypt abscesses prevalence in chronic colitis were higher in animals consuming MP. According to the literature, based on the qualitative morphological assessments and the pathological score (semi-quantitative assessment of the severity of pathological changes), MP consumption leads to a more severe course of both acute and chronic experimental colitis [10,11,12,13,14,15].

Thus, in mice with chronic colitis, MP consumption led to an increase in the endocrine cell number. Colon endocrine cells produce mainly serotonin. Serotonin can contribute to increased inflammatory infiltration and pro-inflammatory activation of immune cells. MP also caused changes in the tight junction expression that regulate epithelial barrier permeability. And, most importantly, the animals that received MP had a more pronounced ulcerative-inflammatory process in the distal colon. Consequently, MP aggravates the chronic experimental colitis course.

## 5. Conclusions

Exposure to 5 μm polystyrene microparticles for 12 weeks at a dose of 1.48 mg/kg/day (2.17 × 10^7^ particles/kg/day) did not cause pathological changes in the mouse colons, but changes in the epithelial barrier were observed, indicating an increase in its permeability. In chronic colitis with and without MP consumption, changes in the colon were similar and were characterized by the ulcer appearance, severe inflammatory infiltration, a decrease in the goblet cell volume fraction and the highly sulfated mucins content in them, and a decrease in the glycocalyx component of mucin 1 mRNA expression. However, the prevalence of pathological changes and the number of chromogranin A-positive endocrine cells were significantly higher in animals consuming MP. Consequently, MP consumption leads to a more severe chronic colitis course. MP is probably one of the factors that provokes exacerbation or the development of a more severe form of colitis in people with IBD.

## Figures and Tables

**Figure 1 toxics-13-00701-f001:**
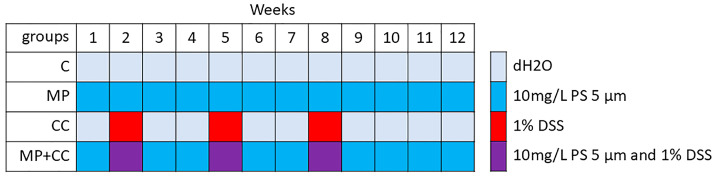
Experimental design.

**Figure 2 toxics-13-00701-f002:**
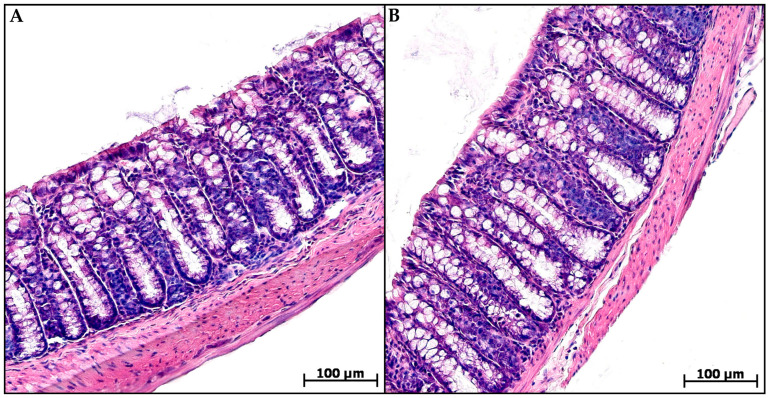
Distal colon of control (**A**) and microplastic treated (**B**) mice, hematoxylin and eosin.

**Figure 3 toxics-13-00701-f003:**
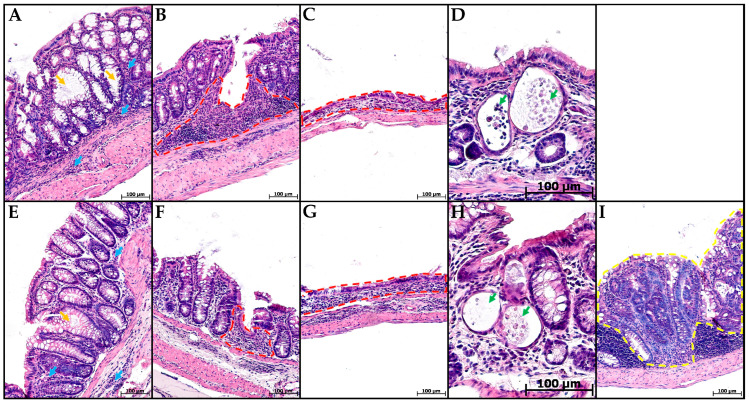
Distal colon of mice with chronic colitis (**A**–**D**) and chronic colitis with microplastic consumption (**E**–**I**), stained with hematoxylin and eosin. (**A**,**E**)—inflammatory infiltration (blue arrows), crypt architecture disorders (orange arrows), (**B**,**F**)—narrow ulcers (red dotted line), (**C**,**G**)—extensive ulcers (red dotted line), (**D**,**H**)—crypt abscesses (green arrows), (**I**)—adenocarcinoma (yellow dotted line).

**Figure 4 toxics-13-00701-f004:**
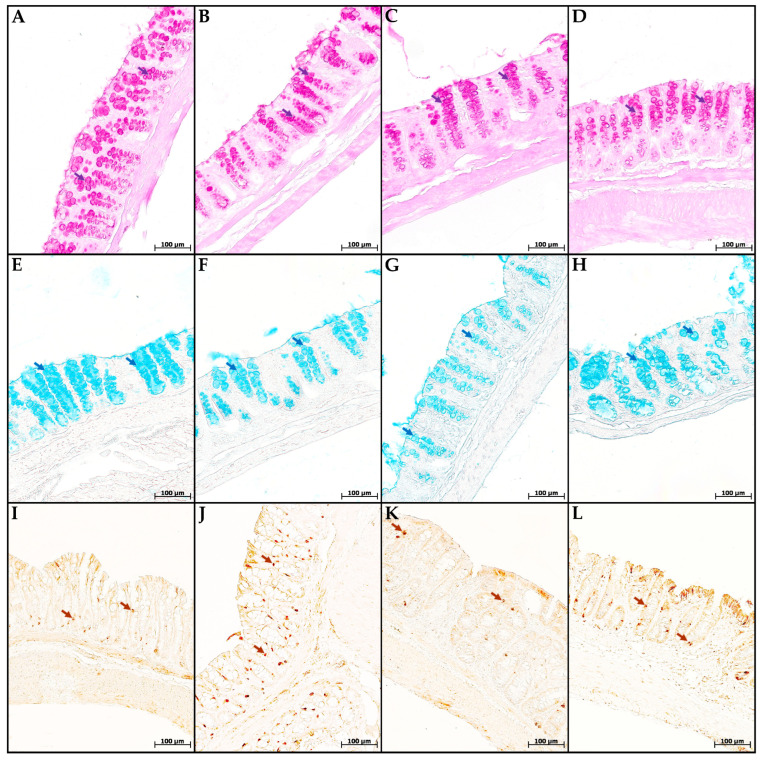
Neutral (**A**–**D**) and highly sulfated (**E**–**H**) mucins in goblet cells and endocrine cells (**I**–**L**) in the distal colon of mice in the control group (**A**,**E**,**I**), fed with microplastics (**B**,**F**,**J**), with chronic colitis (**C**,**G**,**K**) and with chronic colitis with microplastic consumption (**D**,**H**,**L**). (**A**–**D**) PAS reaction, neutral mucins in goblet cells are stained purple (violet arrows); (**E**–**H**) staining with Alcian blue pH 1.0, highly sulfated mucins in goblet cells are stained blue (blue arrows); (**I**–**L**)—immunohistochemical reaction with antibodies to chromogranin A, enteroendocrine cells of dark brown color (brown arrows).

**Figure 5 toxics-13-00701-f005:**
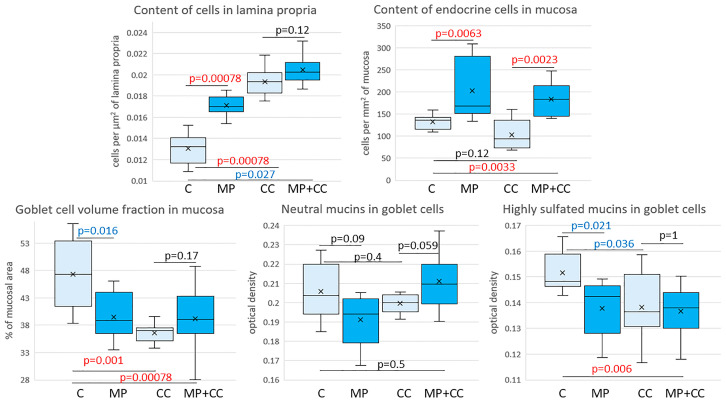
Morphometric study of the intestinal mucosa of mice from the control group (C), fed with microplastics (MPs), with chronic colitis (CC) and with chronic colitis with microplastic consumption (MP+CC) (n = 8). Red numbers are statistically significant changes, blue numbers are trends towards change.

**Figure 6 toxics-13-00701-f006:**
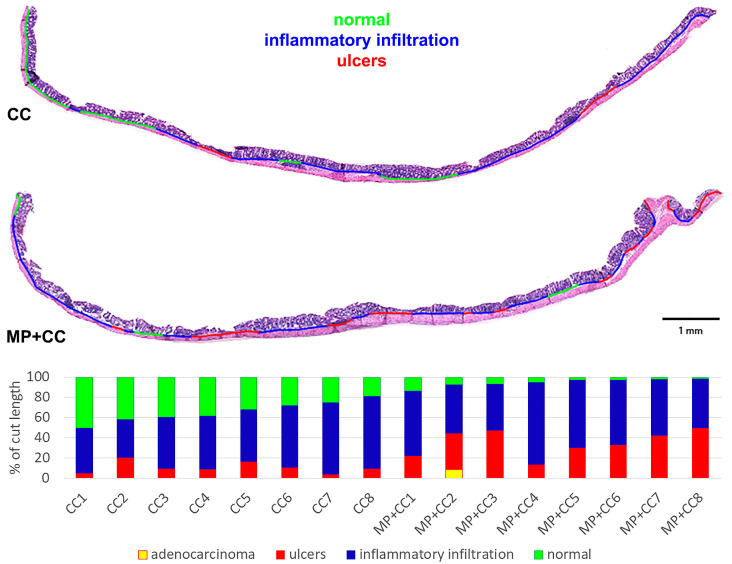
Prevalence of pathological changes in the distal colon during chronic colitis in mice not treated with MP (CC) and treated with MP (MP+CC).

**Figure 7 toxics-13-00701-f007:**
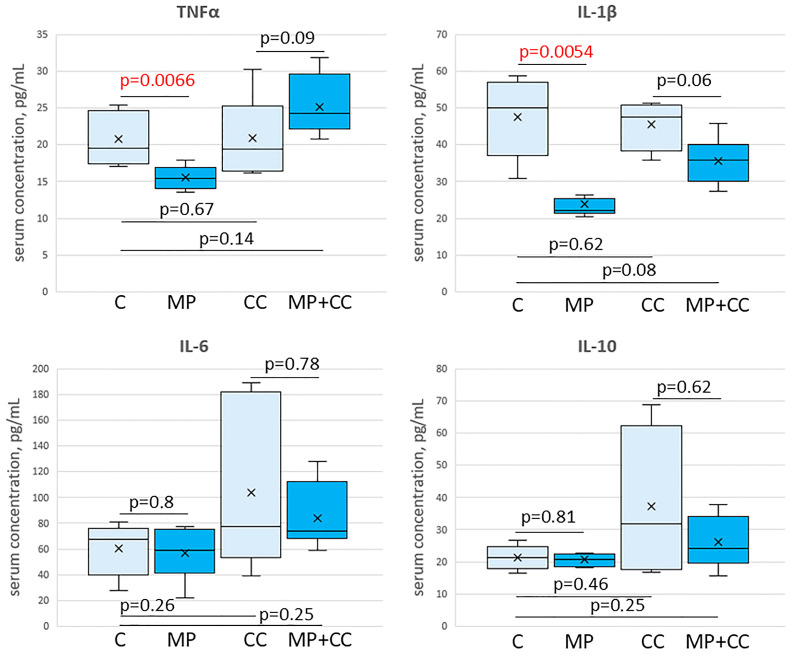
The content of cytokines TNFα, IL-1β, IL-6 and IL-10 in the blood serum of mice of the control group (C), fed with microplastics (MP), with chronic colitis (CC), and with chronic colitis with microplastic consumption (MP+CC) (n = 8). Red numbers are statistically significant changes.

**Table 1 toxics-13-00701-t001:** Prevalence of pathological changes in the distal colon during chronic colitis in mice not treated with MP (CC) and treated with MP (MP+CC), Me (0.25; 0.75), Mann–Whitney U test (n = 8). Red numbers are statistically significant changes.

Group	Prevalence of All Pathological Changes, %	Prevalence of Ulcers, %	Prevalence of Inflammatory Infiltration, %	Number of Crypt Abscesses per Section
CC	65.2 (60.1; 72.9)	9.7 (8.1; 12.4)	53.0 (51.3; 66.1)	0 (0; 1)
MP+CC	96.2 (93.3; 97.8)	34.8 (28.3; 43.7)	55.7 (48.4; 65.6)	1.5 (1; 2)
*p*	0.00078	0.0016	0.6	0.021

**Table 2 toxics-13-00701-t002:** mRNA expression of genes encoding claudins, mucins, proapoptotic factor *Bax* and proliferation marker *Mki67*, relative to expression of *Actb* mRNA in the wall of the medial part of the intestinal tract of mice in the control group (C), fed with microplastics (MP), with chronic colitis (CC) and with chronic colitis with microplastic consumption (MP+CC), Me (0.25; 0.75), Mann–Whitney U test (n = 8). Red numbers are statistically significant changes, blue numbers are trends towards change.

	Groups			
Genes	C	MP	CC	MP+CC
*Muc1*	0.28 (0.11; 0.48)	0.14 (0; 0.18) P_(MP/C)_ = 0.39	0 (0; 0) P_(CC/C)_ = 0.007 ↓↓	0 (0; 0) P_(MP+CC/C)_ = 0.012 ↓ P_(MP+CC/C)_ = 1
*Muc3*	130 (11; 1610)	31 (10; 735) P_(MP/C)_ = 0.87	485 (122; 2594) P_(CC/C)_ = 0.13	574 (502; 2771) P_(MP+CC/C)_ = 0.17 P_(MP+CC/C)_ = 0.53
*Muc13*	1.36 (0.07; 19.15)	0.47 (0.01; 16.08) P_(MP/C)_ = 0.71	2.53 (1.90; 11.64) P_(CC/C)_ = 0.052	6.32 (1.82; 13.92) P_(MP+CC/C)_ = 0.42 P_(MP+CC/C)_ = 0.36
*Cldn2*	0.18 (0.08; 0.27)	0.63 (0.22; 1.74) P_(MP/C)_ = 0.040 ↑	0.26 (0; 0.51) P_(CC/C)_ = 0.71	0 (0; 0.04) P_(MP+CC/C)_ = 0.036 ↓ P_(MP+CC/C)_ = 0.044 ↓
*Cldn4*	1.15 (0.30; 15.02)	0.87 (0.02; 5.42) P_(MP/C)_ = 0.32	2.19 (0.06; 10.54) P_(CC/C)_ = 0.53	10.9 (2.66; 12.74) P_(MP+CC/C)_ = 0.33 P_(MP+CC/C)_ = 0.076
*Cldn7*	0.48 (0.14; 18.00)	0.06 (0; 4.21) P_(MP/C)_ = 0.13	1.92 (0.20; 6.36) P_(CC/C)_ = 0.55	12.52 (5.25; 25.67) P_(MP+CC/C)_ = 0.27 P_(MP+CC/C)_ = 0.026 ↑
*Bax*	1.95 (0.5; 9.09)	1.07 (0.58; 1.07) P_(MP/C)_ = 0.44	3.04 (0; 8.92) P_(CC/C)_ = 0.72	5.63 (0; 12.6) P_(MP+CC/C)_ = 0.95 P_(MP+CC/C)_ = 0.74
*Mki67*	0.22 (0.01; 3.24)	0.3 (0.15; 1.27) P_(MP/C)_ = 0.72	1.54 (0.12; 6.63) P_(CC/C)_ = 0.47	0.14 (0; 5.74) P_(MP+CC/C)_ = 0.75 P_(MP+CC/C)_ = 0.5

## Data Availability

The data that support the findings of this study are available on request from the corresponding author.

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
