# Peer review of "Influence of Microplastics on Manifestations of Experimental Chronic Colitis"

_toxics, 2025, doi:10.3390/toxics13080701_

Round 1

Reviewer 1 Report

Comments and Suggestions for Authors

General comments: This manuscript investigates the impact of microplastic (MP) exposure on the severity of dextran sulfate sodium (DSS)-induced chronic colitis in mice. The topic is timely and addresses a significant environmental health concern. The study design is generally appropriate, combining MP exposure with a well-established colitis model. The authors comprehensively assess histological, cellular, molecular, and inflammatory parameters. The findings suggest that MP exacerbates chronic colitis pathology, particularly ulceration and endocrine cell changes, without inducing standalone pathology in healthy mice. As a researcher in environmental toxicology, I have doubts about the conclusion and need to disclose more details of the experiment.

Specific comments: 

  1. Sample Size and Statistical Power: With n=8 per group and high variability in some parameters (e.g., cytokine levels), the study may be underpowered to detect subtle effects.  Nonparametric tests were appropriately used, but the Bonferroni correction (*p*<0.0085 for 4-group comparisons) is extremely stringent.  The rationale for this threshold should be clarified, and potential Type II errors acknowledged.
  2. Lack of Randomization and Blinding: The explicit statement that "no special techniques were used to randomize animals" and that "all investigators were aware of group allocation" introduces significant risk of bias.
  3. Incomplete Model Characterization: The DSS-only (CC) group showed no significant systemic cytokine elevation vs. controls, suggesting the model was in remission.  While clinically relevant, the absence of active inflammation at endpoint limits interpretation of MP’s impact on acute inflammatory phases.
  4. Figure 4 lacks scale bars and clear annotations for mucin/endocrine cell panels.
  5. General feeling: The authors may have published some papers on this research, so some contents have been briefly expressed in the manuscripts. But this will cause reading difficulties for reviewers and readers. For instance, without a basic introduction to the detection indicators, cell types, and the proteins being detected, people without a medical background cannot understand what enlightenment can be gained from measuring these indicators.
  6. P2, Line 62: Clarify why chronic colitis is "more adequate" for modeling human IBD than acute models.
  7. Specify criteria for "correctly oriented crypts" in morphometry. How were ulcer areas defined?  Include representative images in supplements.
  8. P8, Endocrine Cells: The increase in chromogranin A⁺ cells in MP+CC vs. CC (p<0.05) is a key finding.  Emphasize this in the abstract.
  9. Figure 3: Include arrows/annotations highlighting ulcers, crypt abscesses, and adenocarcinoma.
  10. Due to the differences in structure and surface chemistry, a large number of current studies on microplastics have not distinguished the types of microplastics, thus causing difficulties in toxicology or health risks. Therefore, I suggest that the title keep "Micoplastics", but after the third paragraph of the Introduction section, the focus should be on the PS.
  11. Line 73, As above, it is necessary to clarify the origin, source, particle size, and whether it contains plasticizers and other substances with confirmed biological toxicity of microplastics.
  12. The unit is not standardized (mg/L).
  13. In line 169-180, It is recommended to supplement the 5' and 3' notations for the primer sequence. This part of the content should be presented in table form in the supplementary materials.
  14. In line 193, The formula for real-time quantitative PCR should be independent and embedded in the text, with supplementary explanations provided below the formula. The Actb is not precise. It is recommended to use the target gene. In addition, the formula can be cited to verify the authority of the method.
  15. In Figure 7, The current P-value comparison looks difficult in the graph.  It is suggested to change it to a tree graph comparison method and change the color of the P-values that need to be highlighted.
  16. This study observed no significant pathological changes in the colon of the MP group, but did not provide data on the distribution or excretion of microplastics in the digestive tract. Current research indicates that microplastics can penetrate the intestinal barrier and enter the bloodstream.  However, in this study, the single microplastic treatment group did not produce inflammation.  Does this mean that the microplastics were not absorbed by the intestines?  Should the residue of microplastics in the feces be checked?  Is it possible that microplastics have already been absorbed in the stomach and thus have not caused pathological changes in the colon?  At present, there are many studies on the inflammatory response caused by microplastic exposure.  However, in this study, the typical pro-inflammatory factors TNFα and IL-1β were significantly lower than those in the control group.  What is the reason for this?
  17. In line 414-428, This cannot explain why the two factors (NFα and IL-1β) in this study were significantly lower than those in the control group. I guess there might be some reasons that have been overlooked here and need to be further clarified.

Author Response

Dear Reviewer,

Thank you for your careful review of our manuscript. We have tried to take into account all your comments and answer all your questions.

The English could be improved to more clearly express the research. - We tried to improve English

Are the methods adequately described? Can be improved - We have corrected the materials and methods section, made it more detailed and categorized.

Are the conclusions supported by the results? Can be improved - We have improved the discussion to align the conclusions with the results.

The authors comprehensively assess histological, cellular, molecular, and inflammatory parameters. The findings suggest that MP exacerbates chronic colitis pathology, particularly ulceration and endocrine cell changes, without inducing standalone pathology in healthy mice. As a researcher in environmental toxicology, I have doubts about the conclusion and need to disclose more details of the experiment. - According to our data, microplastics affect the colon of healthy mice: the content of cells in the lamina propria and the number of chromogranin A-positive cells increases, there is a tendency to a decrease in the volume fraction of goblet cells and the content of sulfated mucins in them, a tendency to an increase in the expression of claudin 2 mRNA. However, no pathological changes were detected in the colon: ulcers, erosions, inflammatory infiltrations or edema were absent. And externally, the animals looked healthy: the fur was smooth and shiny, motor activity and body weight did not decrease. Therefore, we consider the observed reactions of the intestinal epithelial barrier to be adaptive. We assume that microplastics have a weak damaging effect on the colon and the defense mechanisms are sufficient to prevent the development of the pathological process. Our assumptions are consistent with the literature data. A number of studies have described the effects of MP such as an increase in the distance between crypts, an increase or decrease in the depth of crypts, an increase in the content of lymphocytes, a decrease in the number of goblet cells in the colon mucosa. But the morphological picture of colitis was not observed in any study. According to 11 studies, MP consumption led to colon mucous membrane and submucosa infiltration by immune cells (Li et al., 2020; Shi et al., 2022; Wen et al., 2022; Xie et al., 2022; Jia et al., 2023; Wang et al., 2023b; Wang et al., 2023a; Huang et al., 2023; Zeng et al., 2024; Sun et al., 2024). But in all cases, immune cells were distributed evenly throughout the connective tissue, there were no violations of the integrity of the epithelium or other signs of inflammation. Therefore, the accumulation of immune cells in the wall of the colon should not be considered a pathological process, but an adaptive reaction: increasing the efficiency of MALT (mucosa-associated lymphoid tissue) in response to increased intestinal permeability. According to many studies, MP exposure increased the proinflammatory cytokines TNFα, IL-1β and IL-6 level, as well as their mRNA expression in colon (Choi et al., 2021a; Liu et al., 2022a; Djouina et al., 2022; Wen et al., 2022; Luo et al., 2022; Xie et al., 2023; Jia et al., 2023; Wang et al., 2023b; Shaoyong et al., 2023; Wang et al., 2023a; Wang et al., 2023c; Zeng et al., 2024; Sun et al., 2024). But, since there are no obvious morphological signs of inflammation in the colon, it is impossible to interpret the local increase in proinflammatory cytokines as the development of pathology. An increase in proinflammatory cytokines is a protective reaction aimed at strengthening the protection of the intestine. And, since there are no signs of colitis, it means that the body copes with the task and the reaction should be considered adaptive. Perhaps with a longer exposure to microplastics or when choosing smaller microplastic particles that more easily overcome the epithelial barrier, a breakdown in adaptation will occur, disadaptation will develop and a transition to a pathological state will occur.

Specific comments:

  1. Sample Size and Statistical Power: With n=8 per group and high variability in some parameters (e.g., cytokine levels), the study may be underpowered to detect subtle effects. Nonparametric tests were appropriately used, but the Bonferroni correction (*p*<0.0085 for 4-group comparisons) is extremely stringent. The rationale for this threshold should be clarified, and potential Type II errors acknowledged. - Each statistical method has its own advantages and disadvantages. As the risk of type II error decreases, the risk of type I error increases. Typically, for multiple comparisons we use built into the STATISTICS program the Kruskal-Wallis test, followed by multiple comparisons of mean ranks for all groups. However, the effects of microplastics are weak and this comparison method did not reveal any changes. Therefore, we chose the more lenient Bonferroni correction. In our work, we performed a comparison between 4 groups of animals, with 6 pairwise comparisons: C vs MP, C vs CC, C vs MP+CC, MP vs CC, MP vs MP+CC, CC vs MP+CC. The Bonferroni correction is calculated by dividing the significance level (α) for one test by the number of tests (n), in our case α=0.05, n=6. This is the reason for the threshold *p*<0.0085 for the Bonferroni correction. We did not include the p-values for the C vs MP+CC and MP vs CC pairs in the text of the article as they had no biological meaning. Pairwise comparisons were performed using the Mann-Whitney U test. Since the samples were small, in order not to lose possible differences due to multiple comparisons, in the case 0.0085<p<0.05, we described the changes as "trends".
  2. Lack of Randomization and Blinding: The explicit statement that "no special techniques were used to randomize animals" and that "all investigators were aware of group allocation" introduces significant risk of bias. - We did not use any special methods of animal randomization, but there was no selection for a certain group based on any characteristics. All animals arrived from the vivarium in one batch in one container. Animals were distributed into cages by technicians who were not aware of the experimental plan. C57BL/6 mice are characterized by high aggression and when animals from different cages are mixed, they often start fighting and kill each other. Therefore, after the initial placement of animals, they were not redistributed into cages. The distribution of cages into groups was also random: technicians placed 4 cages in a row, and we numbered them according to the order of our groups: C, MP, CC, MP+CC. The use of blinding is important when using qualitative or semi-quantitative methods. In our study, all parameters were assessed quantitatively. Morphometric parameters were calculated by two researchers independently and the results were consistent.
  3. Incomplete Model Characterization: The DSS-only (CC) group showed no significant systemic cytokine elevation vs. controls, suggesting the model was in remission. While clinically relevant, the absence of active inflammation at endpoint limits interpretation of MP’s impact on acute inflammatory phases. - We studied the effect of microplastics on the acute colitis model in our previous study (Zolotova, N.; Dzhalilova, D.; Tsvetkov, I.; Makarova, O. Influence of Microplastics on Morphological Manifestations of Experimental Acute Colitis. Toxics 2023, 11, 730. https://doi.org/10.3390/toxics11090730). This time, we reproduced the chronic inflammation model. In addition, there are a number of other studies that investigated the effects of microplastics in acute colon inflammation (Xie et al., 2023; Zheng et al., 2021; Schwarzfischer et al., 2022). Therefore, in this work, we decided to reproduce a colitis model closer to IBD in humans - chronic colitis in the phase of clinical remission. Since the etiology of IBD is unclear, diagnostic and treatment methods that are currently available are not effective enough. Clinical manifestations of IBD are non-specific and include diarrhea, rectal bleeding, fatigue and weight loss. Diagnosis is also complicated by the fact that approximately 25-40% of people with IBD have extraintestinal manifestations such as arthritis, axial spondyloarthritis, uveitis, erythema nodosum and primary sclerosing cholangitis. From the first appearance of IBD symptoms to diagnosis, an average of 2 months to six months passes, and in some cases, diagnosis can take up to 8 years (Cross et al. 2023). Thus, by the time the diagnosis is established, the inflammatory process in the intestines has already reached the chronic phase. The course of both diseases characterized by phases of relapse and remission is unpredictable. Today, IBD is a medically incurable disease and the goal of therapy is to achieve stable remission (Sokic-Milutinovic et al., 2023). However, treatment often only results in the disappearance of clinical symptoms. According to Rosenberg et al. (2013) at endoscopy 45% of patients in clinical remission had endoscopic inflammation. Therefore, the most adequate model of human IBD is experimental chronic colitis in the phase of clinical remission.
  4. Figure 4 lacks scale bars and clear annotations for mucin/endocrine cell panels. - Scale bars are located in the lower right corner of each photo. Arrows have been added to the figure. Explanations have been added to the caption.
  5. General feeling: The authors may have published some papers on this research, so some contents have been briefly expressed in the manuscripts. But this will cause reading difficulties for reviewers and readers. For instance, without a basic introduction to the detection indicators, cell types, and the proteins being detected, people without a medical background cannot understand what enlightenment can be gained from measuring these indicators. - The introduction has been supplemented with a rationale for choosing parameters for analysis:

The most widely used and highly reproducible animal model for studying inflammatory bowel disease is DSS-induced colitis.  This model allows inducing both acute and chronic colitis of varying severity. The DSS-induced colitis model is more representative of Ulcerative Colitis (UC) than Crohn's Disease. Therefore, we used a DSS-induced colitis model and, when choosing parameters to assess the severity of chronic experimental colitis, we relied on studies of UC in humans. The main indicator of the severity of UC in humans is histological changes in the colon [Pavel et al. 2024]. In DSS-induced colitis, to quantify the severity of inflammation in the colon commonly used a Histology score, a semi-quantitative measure derived from ulcers, crypt abscesses and inflammatory infiltration (Li et al., 2020; Wen et al., 2022; Wang et al., 2023; Sun et al., 2024). An important role in the UC development played by disturbances in the colonic epithelial barrier The first component of this barrier is the mucus produced by goblet cells. Mucus forms two layers: an inner dense layer, which is permeable to bacteria and particles larger than 0.5 μm. The second layer is loose and populated by commensal microflora. The main structural component of colon mucus is a highly glycosylated glycoprotein, mucin MUC2. The terminal carbohydrate groups of mucins can be modified by sulfate or sialic acid residues (acidic mucins) or unmodified (neutral mucins). It has been shown in ulcerative colitis, there is a significant reduction in the thickness of the mucus layer and the number of goblet cells, increased amounts of sialomucins and decreased sulfation of mucins. The degree of these changes correlates with the severity of colitis.  In mice knockout for the mucin 2 gene (Muc2–/–), the mucus layer is absent and spontaneous colitis develops, while in heterozygous Muc2+/– mice, DSS-induced colitis is significantly more severe than in wild-type animals. The next component of the epithelial barrier is the glycocalyx, a carbohydrate-rich layer of transmembrane molecules on the apical surface of colonic epithelial cells. In the colon, its primary structural components are membrane-associated mucins. In humans, the major mucins of the colonic glycocalyx are MUC3, MUC12, and MUC17, which share a high degree of structural similarity and are encoded by a single gene (Muc3) in mice. Data on the role of these mucins in the development of ulcerative colitis are contradictory. The glycocalyx of the human colon contains a significant amount of MUC13 mucin. Mice with a knockout of the mucin 13 gene (Muc13–/–) develop more severe DSS-induced colitis than wild-type animals. The abundance of MUC1 in the colonic glycocalyx is not high, but this molecule contains several important signaling sequences. In UC patients MUC1 expression increases, and anti-MUC1 antibodies are found in the blood. In mice knockout for the mucin 1 gene (Muc1–/–), the mucus layer is thicker, and these animals develop DSS-induced colitis with significantly reduced severity compared to wild-type animals. The colonic mucous is lined with a single layer of columnar epithelium. All epithelial cells are formed from stem cells located at the bottom of the crypts. The main types of differentiated epithelial cells are colonocytes, goblet cells, and enteroendocrine cells, each with specialized functions. Colonocytes are the primary absorptive cells, responsible for water and electrolyte absorption. Goblet cells secrete mucus, which lubricates and protects the epithelial lining. Enteroendocrine cells produce and secrete hormones that regulate various digestive processes. The predominant type of enteroendocrine cells in the colon are Ec-cells (enterochromaffin cells) that secrete serotonin. Ulcerative colitis is characterized by a decrease in the number of goblet cells, which correlates with the severity of ulcerative colitis; during remission, the number of goblet cells increases. The number of endocrine cells in the colon and the serotonin content in the tissue decrease during ulcerative colitis exacerbation, and increase during remission and in the long-term course of the disease. Epithelial cells are connected by junctional complexes: desmosomes, adhesive junctions, and tight junctions. For the barrier function of the colon, tight junctions are the most important. They are impermeable to macromolecules and selectively permeable to some ions. Tight junctions are formed by intersecting chains of transmembrane proteins that interact with proteins of the neighboring cell and form a network of point connections between membranes. Claudins are the main structural components of tight junctions. In the colon of patients with ulcerative colitis, the content of claudin 2 protein increases, and the content of mucins 4 and 7 decreases. Knockout of claudins 2, 4, 7 genes in mice results in more severe DSS colitis or development of spontaneous colitis (Zolotova et al., 2019; Camilleri 2023; Neurath et al., 2025).

  1. P2, Line 62: Clarify why chronic colitis is "more adequate" for modeling human IBD than acute models. - Since the etiology of IBD is unclear, diagnostic and treatment methods that are currently available are not effective enough. Clinical manifestations of IBD are non-specific and include diarrhea, rectal bleeding, fatigue and weight loss. Diagnosis is also complicated by the fact that approximately 25-40% of people with IBD have extraintestinal manifestations such as arthritis, axial spondyloarthritis, uveitis, erythema nodosum and primary sclerosing cholangitis. From the first appearance of IBD symptoms to diagnosis, an average of 2 months to six months passes, and in some cases, diagnosis can take up to 8 years (Cross et al. 2023). Thus, by the time the diagnosis is established, the inflammatory process in the intestines has already reached the chronic phase. The course of both diseases characterized by phases of relapse and remission is unpredictable. Today, IBD is a medically incurable disease and the goal of therapy is to achieve stable remission (Sokic-Milutinovic et al., 2023). However, treatment often only results in the disappearance of clinical symptoms. According to Rosenberg et al. (2013) at endoscopy 45% of patients in clinical remission had endoscopic inflammation. Therefore, the most adequate model of human IBD is experimental chronic colitis in the phase of clinical remission.
  2. Specify criteria for "correctly oriented crypts" in morphometry. How were ulcer areas defined? Include representative images in supplements. - Correctly oriented crypts are crypts cut lengthwise from the base of the muscular plate to the opening into the intestinal lumen. Examples of photographs on which the morphometric study was conducted are shown in Figure 4. Ulcers: there are no crypts in the ulcer area, the surface is lined with eosinophilic cuboidal or low prismatic epithelium (actively migrating and proliferating cells) without goblet cells, the bottom of the ulcers is represented by granulation tissue (young connective tissue) is represented by granulation tissue, in which fibroblasts, fibrocytes, lymphocytes and plasma cells were determined. Examples of ulcers are shown in Figure 3 B,C,F,G, the ulcer borders are outlined with a dotted line.
  3. P8, Endocrine Cells: The increase in chromogranin A⁺ cells in MP+CC vs. CC (p<0.05) is a key finding. Emphasize this in the abstract. - Added the result on neuroendocrine cells to the annotation.
  4. Figure 3: Include arrows/annotations highlighting ulcers, crypt abscesses, and adenocarcinoma. - Arrows and borders added.
  5. Due to the differences in structure and surface chemistry, a large number of current studies on microplastics have not distinguished the types of microplastics, thus causing difficulties in toxicology or health risks. Therefore, I suggest that the title keep "Micoplastics", but after the third paragraph of the Introduction section, the focus should be on the PS. - Earlier we conducted a literature review еo assess the MP effect on the healthy mice colon (Zolotova N, Dzhalilova D, Tsvetkov I, Silina M, Fokichev N, Makarova O. Microplastic effects on mouse colon in normal and colitis conditions: A literature review. PeerJ. 2025 Feb 10;13:e18880. doi: 10.7717/peerj.18880. PMID: 39950042; PMCID: PMC11823654.). We examined the PubMed for those articles included queries ``microplastics, colon, mice'', ``microplastics, large intestine, mice'' and ``microplastics, large bowel, mice'' through all the time up to 04/2024 and 34 materials were selected. In 26 studies used polystyrene (PS) particles, 4 studies used polyethylene (PE), single articles used polypropylene (PP), polyvinyl chloride (PVH), polyethylene terephthalate (PET), and low-density polyethylene (LDPE), two studies used biodegradable plastics: PLGA (poly lactic-co-glycolic acid) and PLA (polylactic acid), and one study did not specify the plastic type. Thus, 75% of studies use PS. No reliable qualitative difference in the effects of different types of microplastics was described. Therefore, we did not exclude other types of plastic besides PS from the discussion.
  6. Line 73, As above, it is necessary to clarify the origin, source, particle size, and whether it contains plasticizers and other substances with confirmed biological toxicity of microplastics. - As microplastic we used a commercial suspension «Micro particles based on polystyrene (PS)» purchased from Sigma-Aldrich (79633, Supelco, USA). We added all the information about the particles provided by the manufacturer to the materials and methods: aqueous suspension, 0% cross-linked, concentration 10% (solids), particle size 5 μm std dev <0.1 μm, coeff var <2%, density 1.05 g/cm3. We have no other information about the particles.
  7. The unit is not standardized (mg/L). - We relied on previously published studies on our topic. In them, the concentration of microplastics in suspension is universally expressed in mg/L or μg/L. For the convenience of comparing various studies, we consider it appropriate to leave the unit of measurement mg/L.
  8. In line 169-180, It is recommended to supplement the 5' and 3' notations for the primer sequence. This part of the content should be presented in table form in the supplementary materials. - The primer sequences have been moved to the supplementary materials and the formatting has been corrected.
  9. In line 193, The formula for real-time quantitative PCR should be independent and embedded in the text, with supplementary explanations provided below the formula. The Actb is not precise. It is recommended to use the target gene. In addition, the formula can be cited to verify the authority of the method. - The formula formatting has been corrected. The study by Wang et al. (2015) (https://pubmed.ncbi.nlm.nih.gov/25445497/) showed that various genes, including Actb, are suitable for gene expression analysis by RT-PCR in colorectal tumors. In this work, we conducted a preliminary study (unpublished data) to evaluate the possibility of using Gapdh, B2m, and Actb as reference genes, with the latter showing the greatest stability and being selected as a reference. Actb has been used as a reference gene in studies of gene expression in the colon in a number of studies by other authors (Choi, Y. J., Kim, J. E., Lee, S. J., Gong, J. E., Jin, Y. J., Seo, S., Lee, J. H., & Hwang, D. Y. (2021). Inflammatory response in the mid colon of ICR mice treated with polystyrene microplastics for two weeks. Laboratory animal research37(1), 31. https://doi.org/10.1186/s42826-021-00109-w; Choi, Y. J., Park, J. W., Kim, J. E., Lee, S. J., Gong, J. E., Jung, Y. S., Seo, S., & Hwang, D. Y. (2021). Novel Characterization of Constipation Phenotypes in ICR Mice Orally Administrated with Polystyrene Microplastics. International journal of molecular sciences22(11), 5845. https://doi.org/10.3390/ijms22115845; Liang, B., Zhong, Y., Huang, Y., Lin, X., Liu, J., Lin, L., Hu, M., Jiang, J., Dai, M., Wang, B., Zhang, B., Meng, H., Lelaka, J. J. J., Sui, H., Yang, X., & Huang, Z. (2021). Underestimated health risks: polystyrene micro- and nanoplastics jointly induce intestinal barrier dysfunction by ROS-mediated epithelial cell apoptosis. Particle and fibre toxicology18(1), 20. https://doi.org/10.1186/s12989-021-00414-1; Ma, J., Wan, Y., Song, L., Wang, L., Wang, H., Li, Y., & Huang, D. (2023). Polystyrene nanobeads exacerbate chronic colitis in mice involving in oxidative stress and hepatic lipid metabolism. Particle and fibre toxicology20(1), 49. https://doi.org/10.1186/s12989-023-00560-8)
  10. In Figure 7, The current P-value comparison looks difficult in the graph. It is suggested to change it to a tree graph comparison method and change the color of the P-values that need to be highlighted. - In Figure 7, the graphs are designed exactly the same as in Figure 5. The data are presented as boxplots (median, interquartile range, minimum and maximum). Black lines connect the compared groups, and p-values are signed above them. If the differences are statistically significant, the p-values are red; if 0.0085<p<0.05 ("trend to change"), the p-values are blue. We constantly use this data formatting in our research. We very often encounter the same formatting in the works of other authors. I don't know what the "tree graph comparison method" is.
  11. This study observed no significant pathological changes in the colon of the MP group, but did not provide data on the distribution or excretion of microplastics in the digestive tract. Current research indicates that microplastics can penetrate the intestinal barrier and enter the bloodstream. However, in this study, the single microplastic treatment group did not produce inflammation. Does this mean that the microplastics were not absorbed by the intestines?  Should the residue of microplastics in the feces be checked?  Is it possible that microplastics have already been absorbed in the stomach and thus have not caused pathological changes in the colon?  At present, there are many studies on the inflammatory response caused by microplastic exposure.  However, in this study, the typical pro-inflammatory factors TNFα and IL-1β were significantly lower than those in the control group. What is the reason for this? - We did not investigate the penetration of MP across the intestinal epithelial barrier. In humans it is assumed that MP absorption may only be limited to ≤0.3% [EFSA Panel on Contaminants in the Food Chain (CONTAM). Presence of microplastics and nanoplastics in food, with particular focus on seafood. EFSA J. 2016 Jun 23;14(6):e04501. doi: 10.2903/j.efsa.2016.4501. PMID: 40007823; PMCID: PMC11847996.]. Liang et al. (2021) treated mice with a single oral gavage fluorescent PS particle 0.05, 0.5 or 5 μm-diameter in dose 250 mg/kg and analyzed its distribution in organs after 24 h. The total bioavailability for 0.05, 0.5 or 5 μm particles were 6.16, 1.53 and 0.46%, respectively. The particles 0.05 and 0.5 μm size were found in the spleen, kidneys, heart, liver, lungs, blood, testis and epididymis, brain and thighbone, some individual signs in the muscles and breastbone, some 0.05 μm particles in the ovaries and uterus. The particles 5 μm size were found in the blood only. The total accumulations of 0.05, 0.5 or 5 μm particles in the intestine were 11.41, 13.66 and 3.84%, respectively. The particles 5 μm size accumulated primarily in the stomach and large intestine. In study of Liu et al. (2022) healthy mice and mice with acute colitis were exposed to 5 μm-diameter fluorescent PS particles at concentrations of 500 μg/L for 28 days and then distribution of fluorescent MPs in mice was assessed by using a small animal living imaging system. In the healthy mice a small amount of fluorescence was distributed in the abdomen, whereas a 13 times stronger fluorescence intensity was detected in the intestinal tract of mice with acute colitis. Thus, PS particles 5 μm-diameter are almost not absorbed in the digestive system and they accumulate in the large intestine. In colitis their intestinal absorption increases, probably due to the formation of ulcers and increased gut permeability. We could not find an explanation for the decrease in blood levels of proinflammatory cytokines TNFα and IL-1β after microplastic consumption. This issue requires additional research.
  12. In line 414-428, This cannot explain why the two factors (NFα and IL-1β) in this study were significantly lower than those in the control group. I guess there might be some reasons that have been overlooked here and need to be further clarified. - We could not find an explanation for the decrease in blood levels of proinflammatory cytokines TNFα and IL-1β after microplastic consumption. We found only 5 publications that assessed blood cytokine levels during microplastic consumption. Their data are contradictory and the authors do not explain the obtained result in any way (Li et al. 2020; Liu et al. 2022; Luo et al. 2022; Zheng et al. 2021; Zha et al. 2024). More often, the content of cytokines is assessed not in the blood, but in the tissue of the colon. MP exposure increased the proinflammatory cytokines TNFα, IL-1β and IL-6 level, as well as their mRNA expression (Choi et al., 2021a; Liu et al., 2022a; Djouina et al., 2022; Wen et al., 2022; Luo et al., 2022; Xie et al., 2023; Jia et al., 2023; Wang et al., 2023b; Shaoyong et al., 2023; Wang et al., 2023a; Wang et al., 2023c; Zeng et al., 2024; Sun et al., 2024). However, one study noted a decrease in Il1b mRNA expression (Sun et al., 2021). Regarding the anti-inflammatory cytokine IL-10, the data were contradictory: some studies demonstrated a decrease in its level (Wen et al., 2022; Jia et al., 2023), while others report—an increase in its mRNA expression (Sun et al., 2021; Choi et al., 2021a). There were same data on the IFNγ level increase and the mRNA Ifng, Tgfb, Il8, Il17a, Il22 expression when exposed to MP (Sun et al., 2021; Choi et al., 2021a; Liu et al., 2022a; Djouina et al., 2022; Luo et al., 2022; Zeng et al., 2024). It is possible that the observed decrease in TNFα and IL-1β in the blood with microplastic consumption is associated with the migration of immune cells producing them from the blood to the wall of the colon. However, we cannot provide a reasonable explanation for the observed changes. This issue requires additional research.

Reviewer 2 Report

Comments and Suggestions for Authors

The potential health risks associated with microplastic exposure warrant further investigation. This study aims to evaluate the possible link between microplastics and the development of chronic inflammatory intestinal diseases. However, the experimental design utilized a concentration of 10 mg/L of 5-micron polystyrene particles based on a mouse model, which does not accurately reflect the exposure levels relevant to human chronic colitis. Furthermore, such concentrations are not typically present in human diets, thereby limiting the scientific relevance and applicability of the findings.

Author Response

Thank you for your expert review of our manuscript.

Does the introduction provide sufficient background and include all relevant references? Must be improved - we have expanded the introduction

Is the research design appropriate? Must be improved - We described the experimental plan in more detail, structured the description, and provided a rationale for the chosen plan.

Are the methods adequately described? Must be improved - We have supplemented the introduction with a justification for the selected evaluation parameters. We have described the methods in more detail.

Are the results clearly presented? Can be improved - we have improved the illustrations

Are the conclusions supported by the results? Can be improved- We have improved the discussion to align the conclusions with the results.

Are all figures and tables clear and well-presented? Can be improved - We have added arrows and borders to the drawings and expanded the descriptions below the pictures.

The potential health risks associated with microplastic exposure warrant further investigation. This study aims to evaluate the possible link between microplastics and the development of chronic inflammatory intestinal diseases. However, the experimental design utilized a concentration of 10 mg/L of 5-micron polystyrene particles based on a mouse model, which does not accurately reflect the exposure levels relevant to human chronic colitis. Furthermore, such concentrations are not typically present in human diets, thereby limiting the scientific relevance and applicability of the findings.

Accordingly, the PS dose was approximately 1.48 mg/kg/day, and the DSS dose was 1.48 g/kg/day. The diameter of the polystyrene spheres was 5 μm, the density was 1.05 g/cm3, therefore, the dose in terms of particles was 2.17 x 107 particles/kg/day. Existing methods for identifying MP and determining its concentration in biological samples of humans, animals, and food products are not standardized and not accurate. Estimates of human MP consumption are mostly indirect and based on estimates of MP content in water and food. According to various estimates, human MP intake is from 0.2 to 10.2 mg/kg/day or from 2 to 1.1x106 particles/kg/day [Zolotova, N., Dzhalilova, D., Tsvetkov, I., Silina, M., Fokichev, N., & Makarova, O. (2025). Microplastic effects on mouse colon in normal and colitis conditions: A literature review. PeerJ13, e18880. https://doi.org/10.7717/peerj.18880]. The MP dose in our study corresponds to human microplastic intake assessment of Senathirajah et al., (2021): 0.2–10 mg/kg/day,  and only an order of magnitude higher than the estimate by Liu et al. (2021): 1.1x106 particles/kg/day. In addition, it should be taken into account that the rate of passage of food through the digestive system in mice is approximately 10 times higher than in humans. In addition, mice have a significantly higher metabolic rate than humans. Therefore, the dose used in our study is adequate for human microplastics exposure, at least in plastic-polluted regions.

In addition, the production of plastic products is growing much faster than their recycling, which leads to the accumulation of plastic waste and the formation of secondary microplastics. If no measures are taken to solve this problem, human consumption of microplastics will only increase over the years.

Human MP consumption: 

  • 39,000–52,000 particles/person/year (2 particles per kg of body weight per day) through water and food (Cox et al., 2019); 
  • up to 458,000 particles/person/year  through tap water and 3,569,000 particles/person/year through bottled water (18 particles/kg/day) (Danopoulos, Twiddy & Rotchell, 2020); 
  • over 90,000 particles/person/year (5 particles/kg/day) (Van Raamsdonk et al., 2020); 
  • 1–5 g/person/week (0.2-10.2 mg/kg/day) (Senathirajah et al., 2021); 
  • 9 × 1010particles/person/year (1.1x106 particles/kg/day) (Liu et al., 2021); 
  • 426 µg/kg bw/day for preschoolers (43 mg/kg/day) (Ke et al., 2023);
  • 96 particles/kg/day (average adult daily intake via food and beverages) (Zuri, Karanasiou & Lacorte, 2023).

Reviewer 3 Report

Comments and Suggestions for Authors

This manuscript presents several pieces of scientific evidence from a study evaluating the effects of microplastic (PS) ingestion on the development of chronic experimental colitis in mice. Key findings include: PS exposure did not induce colitis in healthy mice, but significantly worsened symptoms and enlarged lesions in experimental chronic colitis compared to acute colitis. This manuscript requires the following revisions and additions for publication:

- Additional information should be provided regarding how MP homogeneity was maintained in the solution when measuring intake of PS-containing solutions, as well as the presence or absence of unconsumed loss.

- Results during acute colitis differed from those of other researchers. The basis for this discrepancy was insufficient, and the discussion section should be supplemented.

- Transmembrane mucins and Cldn2 expression during MP exposure for 12 weeks or longer were discussed as a result of physical adaptation. A more specific mechanism should be presented for this.

- Materials and Methods should be categorized by experimental method.

- (Mean ± SD) repetition should be omitted.

Comments on the Quality of English Language

Overall English expressions need to be revised and supplemented.

Author Response

Dear Reviewer,

Thank you for your expert review of our manuscript.

The English could be improved to more clearly express the research. - we improved English

Does the introduction provide sufficient background and include all relevant references? Must be improved - we have expanded the introduction

Is the research design appropriate? Can be improved - We improved the rationale and described the experimental design in more detail.

Are the methods adequately described? Must be improved - We have described the methods in more detail and added justification for the selected parameters

Are the results clearly presented? Can be improved - we have improved the presentation of results

Are the conclusions supported by the results? Must be improved - we expanded the discussion

Are all figures and tables clear and well-presented? Must be improved - We have added arrows and borders to the drawings and expanded the descriptions below the pictures.

- Additional information should be provided regarding how MP homogeneity was maintained in the solution when measuring intake of PS-containing solutions, as well as the presence or absence of unconsumed loss. - A glass drinking bowl was filled with 125 ml of distilled water and added 12.5 μm of a stock suspension of microplastics (10% suspension, 100 mg/ml). The drinking bowl was closed and shaken thoroughly by hand. The drinking bowl was installed in the cage. 2-3 times every day the drinking bowl was removed from the cage, shaken by hand to prevent the sedimentation of microplastic particles and installed back in the cages. Twice a week, the suspension in the drinking bowls was replaced. The drinking bowls were removed from the cages, the remaining liquid was poured into a measuring flask and the volume of the remainder was measured (from 10 to 40 ml remained). After measuring the volume, the remaining liquid was poured out, the drinking bowls were thoroughly washed, and a new suspension of microplastics was prepared. During manipulations with the drinkers, several drops of water were lost. According to our rough estimates, the losses were less than 5 ml per drinker per week, and we neglected these losses. From the initial volume of liquid (125 ml), the volume of the remainder (10-40 ml) was subtracted, added for the number of mice in the cage (8 animals) and divided by the number of days of liquid consumption from the drinker (3-4 days), thus, the liquid consumption of the animals was calculated. The consumption of distilled water (group C) and sodium dextran sulfate solution (group CC) was calculated similarly.

- Results during acute colitis differed from those of other researchers. The basis for this discrepancy was insufficient, and the discussion section should be supplemented. - We used a model of chronic colitis, not acute colitis. The model was successfully reproduced. The results we obtained coincide with the literature data on experimental chronic colitis. Although the chronic DSS-induced colitis model is widely used, there is very little information in the literature on changes in the components of the epithelial barrier. In our previous studies, we showed that in mice with chronic colitis induced by one cycle of 1% DSS solution for 5 days, on the 28th day of the experiment, compared with the control, the content of sulfomucins in goblet cells in the distal colon was statistically significantly reduced. The volume fraction of goblet cells, the content of neutral mucins in them, and the number of chromogranin A-positive enteroendocrine cells did not differ from the control. In the medial colon, the expression of Muc1 and Cldn2 mRNA increased, Muc13 decreased, and the expression of Muc3 and Cldn4 mRNA did not differ from the control. (Zolotova N.A. et al., 2016; Zolotova N.A. et al., 2018).

- Transmembrane mucins and Cldn2 expression during MP exposure for 12 weeks or longer were discussed as a result of physical adaptation. A more specific mechanism should be presented for this. - In our experiment, when exposed to microplastics, we only found a trend towards increased claudin 2 mRNA expression. The mechanism of this change has not been described in the literature and this issue was not the goal of our study. It is known that experimental colitis is milder in animals with claudin 2 hyperexpression. Therefore, we consider the increase in claudin 2 expression as an adaptive response. We cannot provide any additional information on this issue. The mechanisms of the influence of microplastics on the expression of mucins and claudins in cells is a separate scientific problem requiring additional research.

- Materials and Methods should be categorized by experimental method. - We have classified the materials and methods.

- (Mean ± SD) repetition should be omitted. - corrected

Round 2

Reviewer 1 Report

Comments and Suggestions for Authors

The revised manuscript is clearer and more complete. The author's responses are sincere and genuine, and some minor issues existing in the original manuscript have been resolved. This is a highly valuable research, which has significant reference value for assessing the risk of current microplastic pollution.

Reviewer 2 Report

Comments and Suggestions for Authors

The author has thoroughly revised the manuscript in accordance with the reviewers' comments and suggestions. The revised version is now deemed suitable for acceptance.

Reviewer 3 Report

Comments and Suggestions for Authors

The manuscript has been appropriately revised and supplemented in accordance with the review criteria.